# Involvement of CB1R and CB2R Ligands in Sleep Disorders and Addictive Behaviors in the Last 25 Years

**DOI:** 10.3390/ph18020266

**Published:** 2025-02-18

**Authors:** Marcel Pérez-Morales, Rodolfo Espinoza-Abad, Fabio García-García

**Affiliations:** 1Health Sciences Department, Metropolitan Autonomous University, Campus Lerma, Lerma de Villada, Mexico City 52000, Estado de Mexico, Mexico; m.perez@correo.ler.uam.mx; 2Health Sciences Graduate Program, Health Sciences Institute, Veracruzana University, Xalapa 91190, Veracruz, Mexico; rodespinoza@uv.mx

**Keywords:** CB1R, CB2R, sleep disorders, psychostimulants, depressants

## Abstract

Over the last three decades, the decriminalization and legalization of therapeutic and recreational marijuana consumption have increased. Consequently, the availability of marijuana-based products associated with its therapeutic use has increased. These developments have stimulated research on cannabinoids involving a wide range of animal models and clinical trials. Also, it is reported that cannabinoids promote sleep in animal models and naïve human participants, and they seem to improve insomnia and sleep apnea in patients. However, evidence from rigorous clinical trials is needed. In addition, among several physiological processes, cannabinoid receptors modulate dopamine synthesis and release. In this regard, the side effects of marijuana and marijuana derivatives must not be ignored. The chronic consumption of marijuana could reduce dopamine responsivity, increase negative emotionality, and induce anhedonia. Research on the neurobiological changes associated with cannabinoid ligands in animal models, in regard to the consumption of both marijuana and marijuana-based compounds, must improve and the effectiveness of the therapeutic outcomes in clinical trials must be guaranteed. In this review, we include a detailed description of the mechanisms of action of cannabinoids on the brain and their impact on sleep disorders and addictive behaviors to emphasize the need to understand the potential risks and benefits of their therapeutic and recreational use. Evidence from basic research and clinical trials from papers published between 2000 and 2024 are included. The pharmacodynamics of these compounds is discussed in terms of sleep–wake regulation, drug addiction, and addictive behaviors.

## 1. Introduction

Human marijuana consumption dates back thousands of years, with records from ancient cultures in Central Asia being the first evidence of its use. During this time, it was employed for distinct purposes, including for religious reasons or rituals, recreational and therapeutic activities [1,2], to produce hemp paper and textile fibers, and to treat physical and mental disorders [3]. In 1925, the League of Nations signed the Opium Convention, adding, for the first time, cannabis to the list of drugs subject to international control [4]. In the 1960s, along with the struggles for civil rights, sexual and gender liberation, and the hippie counterculture movement, which proclaimed free love, peace, and the rejection of both the Vietnam War and the traditional values of society, its consumption as a psychotropic drug in the United States and in several countries in the Western world, started to become popular [5,6]. Pop culture and psychedelic rock also reinforced this popularization. The Beatles, The Rolling Stones, and Bob Dylan, for instance, wrote lyrics about marijuana or were recognized to be marijuana users, which influenced the consumption of marijuana among teenagers and young adults at the time [7]. Movies and literature of the 1960s also contributed to the conceptualization of marijuana as a symbol of resistance and rebellion against the authorities [8,9]. In the 1970s, the Controlled Substances Act classified marijuana as a Schedule I drug, at the same level as heroin, making it illegal to possess, use, cultivate, buy, or sell in the United States [10]. This also implied that the medical use of marijuana was not acceptable and that it had the highest potential for misuse. However, towards the end of the 1970s, some US States started to decriminalize the possession of small quantities of marijuana. Throughout the 1980s and 1990s, marijuana consumption continued to be popular, despite the war against drugs promoted by the Reagan administration. Although Mikuriya and Aldrich pointed out the dangers in terms of the availability of highly potent varieties of marijuana in the late 1980s [11], since the 2000s, a global movement in favor of the decriminalization and legalization of both the recreational and therapeutic use of marijuana has led to a shift in public perceptions. Consequently, the availability of marijuana-based products has increased, reinforcing the status of marijuana as the most consumed illicit drug in the world. Although marijuana and marijuana-based products seem to improve sleep apnea and insomnia, they also interact with the brain’s reward system, and their side effects should be studied carefully to guarantee their effectiveness as therapeutic tools.

## 2. Search Strategy

An initial search for the terms “marijuana”, “sleep disorders”, and “addiction” using PubMed, which was available at the Metropolitan Autonomous University, to visualize the state of the art in terms of these topics, yielded a total of 132 papers, published between 1902 and 2024. To identify the literature to be included in this narrative review, another search was undertaken using several electronic medical databases (PubMed, Science Direct, Scopus, Web of Science), using the terms “marihuana/marijuana”, “cannabis”, “therapeutical cannabis”, “cannabinoids”, “cannabinoid receptor”, “THC”, “CBD”, “synthetic cannabinoids”, “substance use disorder” (SUD), and “sleep” and “sleep disorders” (“insomnia”, “sleep apnea”, “narcolepsy”, “restless leg syndrome”, “REM sleep behavior disorder”), and “addiction” (“addictive behaviors”, “brain reward system”, “psychostimulants of the central nervous system”, “depressants of the central nervous system”), in regard to both preclinical research papers and clinical trials, between October and December 2024. This literature search contained “books and documents”, “clinical trials”, “meta-analyses”, “randomized controlled trials”, “reviews”, and “systematic reviews”. The paper inclusion criteria were peer-reviewed original research, peer-reviewed reviews, cannabinoid therapeutic administration in humans, and cannabinoid experimental administration in animal models, published between 2000 and 2024.

## 3. Phytocannabinoids, Endocannabinoids, and Synthetic Cannabinoids

Currently, there are more than 500 chemical compounds and more than 100 cannabinoids in the marijuana plant; delta-9-tetrahydrocannabinol (THC) and cannabidiol (CBD) have been identified as the main bioactive compounds [12,13]. The effects of marijuana consumption include an increased heart rate, a decrease in body temperature, an increase in appetite and sleep, an alteration in the perception of time, which may be perceived as shorter or longer than it is, and an increase in the vividness of perceived stimulus [14,15]. While THC is responsible for these intoxicating effects, CBD does not possess psychotropic effects. CBD has also recently been found to exert beneficial effects in animal models of epilepsy, cardiovascular disease, cancer, and inflammation [16].

The effects of cannabinoids are mainly mediated by activating cannabinoid receptor type 1 (CB1R) and cannabinoid receptor type 2 (CB2R), which belong to the superfamily of G protein-coupled receptors. These receptors are expressed in mammals, birds, reptiles, and fishes [17]. The stimulation of these receptors is the result of the majority of the actions of THC [18]. CB1R is expressed in the central nervous system (CNS) and the periphery; within a neuron, it is often localized in axon terminals, and its activation induces the inhibition of neurotransmitter release, including glutamate and GABA, via a presynaptic mechanism [19]. CB2R has been found in the immune system [20,21], but there is evidence of its expression in the brain stem [22], and evidence of its overexpression in the brains of macaques with Simian immunodeficiency virus-induced encephalitis [23], and in the postmortem brains of patients with Alzheimer’s disease [24]. Due to their differential expression, CB1R agonists are strongly associated with psychotropic effects, and increasing interest in cannabinoid agonists and antagonists as alternative therapeutic compounds has been recently directed toward CB2R. CBD, the other main bioactive compound in marijuana, binds poorly to CB1R, and it has been suggested that it acts as a negative allosteric modulator of CB1R [25,26], and that it interacts with the transient receptor potential channels (TRPAs), GPR18, GPR55, and 5-hydroxytryptamine receptor 1A [13], amongst others.

In addition to the phytocannabinoids present in marijuana, the brain synthesizes its endogenous cannabinoid molecules. These endogenous molecules, including enzymes involved in the synthesis and degradation of cannabinoids, led to the discovery of the endocannabinoid system. *N*-arachidonoylethanolamine (anandamide), a CB1R agonist and the natural ligand of transient receptor potential cation channel subfamily vanilloid member 1 (TRPV1) channels and 2-arachidonylglycerol (2-AG), an agonist of both CB1R and CB2R, are the two major cannabinoid ligands that have been isolated from the brain [13].

The discovery of THC, CBD, and endocannabinoids led to the design of synthetic cannabinoids. These compounds, originally developed as potential therapeutics in terms of the endocannabinoid system [27], were synthesized in clandestine laboratories and marketed as “legal cannabis” in the early 2000s, using brand names such as “K2” (North America) and “Spice” (Europe) [28]. These synthetic cannabinoids, such as JWH-018, JWH-073, HU-210, and CP-47,497 [29], are usually soaked and sprayed on dried plant materials and then given the name “herbal highs”, in part due to their ability to avoid detection using standard cannabis screening processes [30]. Synthetic cannabinoids have been found to exhibit greater affinity to CB1R than THC and greater affinity to CB1R than to CB2R; pharmacologically, a series of in vivo and in vitro studies has shown that they are 2–100 times more potent than THC [31] and that they exert THC-like physiological and psychotropic effects [32], but with remarkable potency and efficacy, which often result in serious adverse effects that require medical attention, including psychiatric, cardiovascular, and gastrointestinal sequelae [33]; psychosis, paranoia, anxiety, and confusion [34,35]; and even death [36].

## 4. Cannabinoid Receptors

The cannabinoid system acts as a complex neuromodulator of homeostasis, with retrograde signaling, and it is involved in CNS disorders [37] and the gut–brain axis [38]. Cannabinoid receptors are widely distributed throughout the CNS and are involved in almost every physiological process, including food intake, memory, sleep–wake behavior, and addictive behaviors. Therefore, exogenous, endogenous, and synthetic ligands of CB1R and CB2R offer an alternative approach to treating sleep disorders and addictive behaviors. They have stimulated research involving preclinical and clinical trials in the last three decades, but they must be employed carefully. This review will focus on their involvement in the abovementioned brain circuits of the sleep–wake cycle and the brain’s reward system.

Preclinical research has accounted for the involvement of hypothalamic, mesocortical, and mesostriatal CB1Rs in regulating the sleep–wake cycle, food intake, and addictive behaviors. In this sense, cannabinoid compounds, including phytocannabinoids and endocannabinoids, synthetic cannabinoids, and cannabinoid-based compounds, are being studied and recognized as alternative therapeutics for sleep disorders and addictive behaviors.

### CB1R/CB2R Ligands

When a potential ligand of cannabinoid receptors is discovered, these chemical compounds are evaluated using a rodent tetrad assay, a set of in vivo behavioral and physiological tests sensitive to the effects of cannabinoids [39]. If these compounds provoke hypokinesia, catalepsy, hypothermia, and antinociception in animal models, they are generally classified as agonists of cannabinoid receptors. Since endocannabinoids, the endogenous ligands of cannabinoid receptors, interact at higher concentrations not only with CB1R and CB2R, but also with TRPV1 channels [40], the peroxisome proliferator-activated receptor α (PPARα) [41], and numerous other receptors, including GPR18 and GPR55 [42], this classification is wide. These chemical compounds can be classified as inverse agonists, partial agonists, agonists, neutral antagonists, or antagonists. In the following sections, based on our search strategy, we will discuss their roles in sleep regulation and addictive behaviors as outlined by the results from preclinical studies and clinical trials.

## 5. Sleep–Wake Cycle

Although sleep functions in mammals have been extensively discussed in several papers, clinical observations, autopsy findings in patients with encephalitis lethargica, and research on animal models have allowed us to understand the brain mechanisms of the sleep–wake cycle. In the first part of the 20th century, von Economo, based on their studies on encephalitis lethargica, postulated that the hypothalamus was a complex center of sleep and wakefulness and that it also may be involved in the pathogenesis of narcolepsy [43]; decades later, Moruzzi and Magoun, employing the encéphale isolé and cerveau isolé preparations by Frédéric Bremer using cat brains, at a time when sleep was still considered a state associated with a lack of stimulation, proposed the ascending reticular activating system (ARAS), a network of neurons in the brain stem responsible for orchestrating the transition from wakefulness to sleep [44]. In 1953, Aserinsky and Kleitman described rapid eye movement (REM) sleep in humans [45]. Due to the single-unit activity recording studies carried on cats and rodents in the 1980s and 1990s, we now know that sleep–wake regulation is more sophisticated than the proposed ARAS, that cholinergic neurons in the brain stem are discharged during both wakefulness and REM sleep, that noradrenergic neurons in the locus coeruleus are discharged during wakefulness, that GABAergic neurons in the preoptic hypothalamus are discharged during non-rapid eye movement (NREM) sleep, and that histaminergic hypothalamic tuberomammillary nucleus neurons are discharged during wakefulness [Figure 1] [46]. Also, we know that lateral hypothalamic neurons synthesize orexin/hypocretin (ox/hcrt) or melanin-concentrating hormone (MCH) peptides and that a lack of ox/hcrt is associated with narcolepsy with cataplexy in rodents, dogs, and humans, that MCH neurons are involved in the generation of REM sleep, and that all these brain structures and neurotransmitters and neuropeptides are interconnected in order to regulate the sleep–wake cycle [Figure 1] [47]. CB1R and CB2R are expressed in brain regions that modulate the sleep–wake cycle and interact with these neurotransmitters and neuropeptides. Therefore, cannabinoid ligands have been identified as an alternative therapy to treat sleep disorders.

### Sleep Disorders

Medical cannabis use is often associated with sleep disturbances, and its consumption seems to improve sleep quality in general, and to ameliorate insomnia and sleep apnea in particular. Nevertheless, systematic research in which the effectiveness of cannabinoid ligands is evaluated in patients with sleep disorders is scarce, or studies are limited due to small sample sizes, a high risk of bias, or a lack of experimental controls [49]. For instance, methodologically heterogeneous studies, lacking objective sleep measures, from the 1970s and 1980s, evaluated the potential impact of THC and CBD on sleep and reported that CBD increases sleep [50] and that THC decreases the time it takes to fall asleep [51] in patients with insomnia. Also, although several papers have documented the close relationship between sleep and the endocannabinoid system in humans, this system is so complex that the results in robustly designed studies are sometimes somewhat contradictory. For instance, in sleep-restricted healthy young adults, the 2-AG serum levels were increased [52], but not that of anandamide [53]. These results imply that 2-AG, but not anandamide, regulates the sleep–wake cycle. Nonetheless, the evidence is more consistent in animal models. Whereas the acute intraperitoneal (IP) administration of CBD to rats has been shown to increase NREM sleep and decrease wakefulness [54], the IP administration of SR141716A, a CB1R antagonist, increases wakefulness and reduces sleep [55], suggesting that exogenous and endogenous cannabinoids promote sleep. The intracerebroventricular (ICV) and intrapendunculopontine tegmentum nucleus [56] and hippocampal [57] administration of anandamide or the administration of 2-AG into the lateral hypothalamus [58] increase sleep in rats. Also, in an animal model of maternal separation, which exhibited reduced sleep and increased wakefulness, the administration of 2-AG into the lateral hypothalamus improved this insomniac-like phenotype [59]. These results account for the therapeutic use of CB1R/CB2R ligands to treat poor sleep quality.

Although the literature on the effects of the administration of CB1R/CB2R ligands on the sleep–wake cycle in humans and rodents is abundant, the outcomes are difficult to interpret. In summary, in rodents, CB1R agonists reduce wakefulness and increase both NREM and REM sleep, whereas CB1R antagonists increase wakefulness and reduce both NREM and REM sleep [60]. Although pharmacokinetics, pharmacodynamics, and the route of administration of these compounds vary across studies, these hypnotic effects seem to be mediated by CB1R activation in brain structures involved in the regulation of the sleep–wake cycle through the modulation of ox/hcrt and MCH neurons in the lateral hypothalamus [59]. Concerning human studies, although it has been reported that marijuana constituents improve insomnia, obstructive sleep apnea, REM sleep behavior disorder, and restless legs syndrome [61], the methodologies used are heterogeneous, which makes it difficult to interpret the results. On the one hand, some studies have employed objective measures, including polysomnography and actigraphy, or subjective measures, including questionnaires, such as the Pittsburgh Sleep Quality Index and the Epworth Sleepiness Scale, or a combination of both types of measures. Conversely, some studies have recruited abstinent patients with a history of marijuana consumption or have administered different CB1R ligands, including THC, CBD, and diverse laboratory-developed chemical compounds based on marijuana, to naïve volunteers [62]. Thus, THC increases [63] or reduces [64] REM sleep, increases [65] or reduces NREM sleep [64,66,67], or does not induce any changes to sleep architecture. A summary of CB1R and CB2R ligands, namely antagonists ABD459, SR141716A, and LY320, 135; CB1R agonists (WIN 55, 212-2), synthetic THC (Dronabinol), and CB1R (AM251) or CB2R antagonists (AM630), used in the treatment of sleep disorders, can be found in Table 1.

## 6. Addictive Behaviors

The term “addiction” has been substituted for “substance use disorder” (SUD) in the DSM-5 classification, and it refers to a pattern of symptoms that reflect the use of a substance despite its adverse effects on the consumer; the classification includes a list of 11 criteria for SUDs. Although the DSM-5 classification consists of an “internet gaming disorder”, and it is recommended that further research is required and does not recognize “work addiction” nor “compulsive sexual behavior” as mental health conditions per se, we will include evidence of CB1/CB2 ligands related to these topics [79,80]. We will also use the neurobiological model of addiction proposed by Koob and Volkow as a reference [81]. In this model, addiction is a “chronically relapsing disorder” that involves neurobiological changes associated with compulsive drug seeking and intake, a loss of control in limiting intake, and an emotionally negative state in the absence of the drug. In regard to this neurobiological model, the authors propose “binge–intoxication”, “withdrawal–negative effect”, and “preoccupation–anticipation” as the three stages in the cycle of addiction, and the associated disruptions in three major brain circuits, i.e., the basal ganglia, the extended amygdala, and the prefrontal cortex (PFC), respectively [81]. Whereas this model has been employed to describe cannabis use disorder [82], we will focus on evidence from animal models of addiction, including the paradigms of place preference conditioning and self-administration, and evidence from clinical trials in which CB1/CB2 ligands induce addition reinstatement or prevent relapse in regard to addictive behaviors.

### 6.1. The Brain’s Reward System

Research on the relationship between marijuana and the brain’s reward system is wide ranging. This system, discovered by serendipity, by Olds and Milner in 1953 [83], is essential for survival. Generally, it is a sophisticated and highly regulated system that allows living beings to experience pleasure and create expectations around the events that guarantee survival, like food and water intake and sexual reproduction, by reinforcing these behaviors through the release of diverse neurotransmitters. Nonetheless, the brain’s reward system is flexible and susceptible to learning. Thus, it can be shaped to experience pleasure as a result of drugs or addictive behaviors, and it can create expectations around drug consumption and the performance of addictive behaviors. The brain’s reward system comprises several subcortical and cortical regions, involving various neurotransmitters and ionotropic and metabotropic receptors. Briefly, when, for instance, an animal subjected to fasting is allowed to eat, in parallel to the highly conserved hypothalamic circuitry involved in food intake, dopaminergic neurons in the ventral tegmental area (VTA) are discharged and release dopamine into the Nucleus Accumbens Shell (NAcS) and reinforce food intake; if the animal subjected to fasting is faced with a dangerous stimulus while eating, the amygdala, another component of this system, interrupts the communication between the VTA and the NAcS. The brain’s reward system also comprises the PFC, which allows the animal to evaluate the potential consequences of performing a reinforcing behavior [84], and both the lateral hypothalamus [85] and the ventral pallidum [86].

### 6.2. CB1R/CB2R Ligands and the Brain’s Reward System

While drug addiction is a chronic relapsing sickness, a patient or an animal in withdrawal may relapse at any time [87]. In this context, the effects of THC on the dopaminergic system have been extensively documented [88]. Microdialysis experiments have shown an increase in extracellular dopamine release in the NAcS following IP administration of THC [89], an effect also provoked by the intravenous (IV) administration of highly addictive stimulants and depressants of the CNS, such as cocaine and morphine, respectively [90], suggesting that marijuana interacts with the brain’s reward system in the same way as highly addictive drugs do. In the same way, low doses of THC accelerate dopamine synthesis and release [91], whereas high doses of THC reduce dopamine synthesis [92] in rodent synaptosomes. In vitro whole-cell patch-clamp experiments have shown that dopamine-1 (D1R) and dopamine-2 type receptor (D2R) agonists in combination enhanced firing in the Nucleus Accumbens Core (NAcC), that the enhanced firing of these neurons requires endocannabinoid synthesis and CB1R activation, as compared with the separate stimulation of D1R or D2R agonists alone [93]. Likewise, in a sample of 24 marijuana abusers with a history of smoking four marijuana joints per day/5 days per week for the last 10 years and since the age of 15, Volkow et al. [94] found an increase in negative emotionality, a decrease in positive emotionality, decreased dopamine reactivity to dopaminergic agonists, and reduced dopamine release in the NAcC, as compared to healthy age-matched controls.

Although these results are not easily transposed to humans that sporadically consume marijuana or marijuana constituents or derivatives, considering the importance of the brain’s reward system and the role of dopamine in diverse physiological processes, they highlight the potential risk posed by these compounds to human health. Based on the addictive potential of cannabinoids due to their interaction with the brain’s reward system and based on the theoretical model of addiction proposed by Koob and Volkow [81], we will focus on experiments and clinical trials, where applicable, in which CB1R/CB2R antagonists have been employed to prevent relapse/reinstatement of psychostimulants (cocaine, amphetamines, and methamphetamines) and depressants (opioids) involving the CNS. We will focus on the conditioning place preference (CPP) and self-administration paradigms in animal models. Even though, according to the incentive sensitization theory of addiction, these paradigms assess “liking” and “wanting” [95], respectively, they also allow us to evaluate relapse/reinstatement. “Liking” refers to the pleasurable experience of reward consumption, is not dependent on dopamine, and involves fragile neural systems; “wanting,” or incentive salience, is a form of motivation and involves robust neural systems, including mesolimbic dopamine [96].

### 6.3. Psychostimulants and the Central Nervous System

Although cocaine, amphetamines, and methamphetamines (METs) have therapeutic uses, the use of psychostimulants is second only to cannabis, the most widely abused illicit drug in the world [97,98,99]. They mimic the peripheral actions of norepinephrine on the autonomic nervous system, induce behavioral activation, and increase alertness, arousal, and motor activity; based on their neuropharmacological mechanisms of action, i.e., the indirect activation of monoaminergic postsynaptic receptors, they are classified as indirect sympathomimetics [100].

Animal studies have shown that the IP administration of CBD for 10 consecutive days impairs cocaine-induced CPP and attenuates cocaine self-administration [101]. In regard to the self-administration paradigm, CBD also attenuates cue-induced reinstatement of cocaine-seeking behavior and enhances the effects of stress-induced cocaine reinstatement. These effects are blocked by AM4113, a CB1R antagonist [102], in male CD-1 mice. Regarding the CPP procedure, ICV CBD administration prevents both acquisition- and MET-induced CPP [103].

A recent systematic review has acknowledged only 39 preclinical studies in which CB1R antagonists and some CB2R antagonists were found suppress the reinstatement of cocaine and amphetamine use in animal models of CPP and self-administration [104]. In general, agonists of CB1R enhance reward-seeking behavior, and CB1R antagonists block cocaine-induced CPP and cocaine-induced reinstatement. For instance, AM251, a CB1R/CB2R antagonist (0.1–1 mg/kg), and SR144528, a CB2R antagonist (0.1–1 mg/kg), decrease discrete cocaine-induced reinstatement, but only AM251 (0.3–1 mg/kg) decreases discrete cue-induced reinstatement, in regard to the self-administration paradigm in rats [105]. Xie2-64, a CB2R antagonist, also reduces cocaine self-administration and reduces dopamine extracellular levels in the nucleus accumbens [106]. However, the modulation of cocaine addiction seems to involve the intricate modulation of CB2R. Systemic, intra-nasal, or local intra-accumbens administration of the CB2R agonists, JWH133 and GW405833, inhibit the intravenous self-administration of cocaine and reduce cocaine-enhanced accumben extracellular dopamine in both wild-type and CB1R-knockout mice. In the same study, AM630, a CB2R antagonist, blocked the effects induced by these CB2 agonists [107]. In regard to the same paradigm, systemic, intra-NAcC, and intra-prelimbic cortex (PrC) administration of HU210, a CB1R agonist, increased the lever responses to MET and reinstated both MET cue-induced and priming-induced reinstatement, whereas the administration of AM251 into the regions mentioned above attenuated these effects in male Wistar rats [108]. Likewise, AM251 (0.3–10 mg/kg) blocks the cocaine-induced acquisition of CPP, and JWH133 (1–10 mg/kg), a CB2R agonist, also replicates this effect. In the same study, AM630, a CB2R antagonist, reversed the effects of JWH133, and both AM251 and AM630 prevented hippocampal neuronal activation in CPP-exposed Swiss mice [109]. Pretreatment with JWH133 also blocked cocaine-induced CPP in WT mice, but the deletion of the CB2R in dopamine neurons did not affect cocaine- and amphetamine- nor MET-induced CPP [110]. These studies suggest that CB1R and CB2R modulate, in opposite ways, psychostimulant-induced addictive behaviors. However, more studies on cannabinoid-based therapies in patients with a dependence or experiencing withdrawal from psychostimulants are still required.

### 6.4. Depressants of the Central Nervous System

Depressants of the CNS reduce neuronal activity, whether through the direct or indirect stimulation of GABAergic receptors, through the direct or indirect inhibition of glutamatergic receptors, or through their interaction with opioidergic receptors [111]. Alcohol, benzodiazepines, and opioids fall into this classification, but we will focus on opioids. All these drugs, except for alcohol, have been employed for therapeutic purposes, including pain relief and somnolence, but they also have addictive potential. In this way, a relationship between opioids and cannabinoids has been described. Whereas THC exposure in adolescence has been shown to increase heroin self-administration and increase priming-induced reinstatement, in adulthood, in rats genetically prone to addiction [112], other animal studies have shown that SR141716A, a CB1R antagonist, reduces heroin-seeking behavior [113] and cue-induced reinstatement of heroin, in regard to the self-administration paradigm [114]. Nevertheless, acute subcutaneous THC administration reduces heroin self-administration in rhesus monkeys [115]. Likewise, GAT358, a CB1R antagonist, reduces morphine and relapse to morphine self-administration in mice [116]. Moreover, 14 days of pre-exposure to CP55940, a potent CB1R agonist, increased morphine self-administration in rats [117].

Although cannabis consumption is frequent in opioid-dependent individuals [118,119], there is one study, based on a Medical Outcomes Survey, which involves a self-reported survey to assess health functioning, that found improvements in health and mental functioning and reduced opioid consumption associated with medical cannabis use, in a sample of 2183 participants between the ages of 20 and 70 years, between the ages of 20 and 70 years old [120]. Scavone et al. [121], in a review on opiate dependence and withdrawal, point out the importance of the molecular effects of cannabinoids, interacting with kappa, delta, and mu-opioid receptors in the locus coeruleus–norepinephrine system, which is implicated in regard to the negative consequences of opiate addiction. Given that abstinence from opiates includes negative emotional, aversive physical, and flu-like symptoms, such as restlessness, severe muscle and bone pain, and sleep problems [122,123], cannabinoid agonists, which promote sleep, seem an appropriate alternative to treat opiate dependence and withdrawal. In support of this alternative, cannabinoid agonists may attenuate opioid withdrawal symptoms via the enhancement of endogenous opioid signaling [124,125,126]. However, interactions demonstrating cross-tolerance, mutual potentiation, and receptor crosstalk between opioids and cannabinoids have been shown [127,128]. Thus, additional research on the paradigms of addictive behaviors using animal models and clinical trials is required. A summary of the CB1R/CB2R ligands, including AM4113 and PIMSR (CB1R antagonists); APD 371 (CB2R agonists); and AM1241, AM1710, and LY828360 (CB2R agonists); employed to treat addictive behaviors is provided in Table 2.

## 7. Discussion

Since the late 1990s, animal studies have consistently shown the sleep-promoting effects of cannabinoids. Endogenous ligands of CB1R/CB2R, like anandamide and 2-AG, and synthetic cannabinoids, increase sleep, whereas cannabinoid antagonists of CB1R, like SR141716, decrease sleep, acting on brain regions involved in sleep–wake regulation. The literature on the efficacy of cannabinoid treatments in patients with sleep disorders is somewhat inconsistent due to methodological diversity. For example, there is one randomized, parallel-designed, double-blinded, controlled, and entirely virtual study in which a 5-week long-term supply of CBD capsules (15 mg), according to authors, was found to be safe and to improve sleep in a sample of 1793 adults with sleep disturbances; they employed the online survey Patient-Reported Outcomes Measurement Information System (PROMIS™) Sleep Disturbance SF 8A, to evaluate sleep [140]. In contrast, Suraev et al. [141], in a systematic review, focused on 14 preclinical and 12 clinical studies and concluded that the evidence in favor of cannabinoids to treat sleep disorders is promising, but that it may help to carry out randomized controlled trials on patients with sleep apnea, narcolepsy, insomnia, post-traumatic stress disorder-related nightmares, restless legs syndrome, and REM sleep behavior disorder. However, they emphasize that, at present, due to a lack of rigorously controlled published papers evaluating the safety and efficacy of long-term cannabinoid-based therapies and due to the dangerous moderate-to-high bias in the existent studies, there is not enough evidence to recommend the systematic clinical use of cannabinoids to treat sleep disorders.

Along with the animal studies suggesting that cannabinoid antagonists, such as SR141716, have the potential to reduce drug-seeking responses, i.e., cravings and relapse, by inhibiting the rewarding properties of drugs and by impairing the reconsolidation of drug-rewarding memories, in regard to the treatment of SUD [142], it has been shown that CB1R agonists enhance psychostimulant-seeking behavior and induce relapse in regard to the self-administration paradigm, and also that CB1R antagonists attenuate psychostimulant-seeking behavior, psychostimulant-induced CPP, and block cue-induced and priming-induced relapse. Yet, the evidence on the impact of CB2R agonists and antagonists on psychostimulant addictive behaviors is more complicated. Although the CB2R agonists, JWH133 and GW405833, reduce the self-administration of cocaine and reduce dopamine extracellular release in accumbens [107], it is still a matter of debate whether CB2R is detected in the brain. Some authors have shown that CB2R is overexpressed in the brains of macaques with Simian immunodeficiency virus-induced encephalitis [23] and have accounted for the existence of these receptors in brain regions associated with addictive behaviors [143], suggesting that CB2R is closely related to brain diseases, that it is involved in the effects of abused drugs and, mainly, that it is modulated after the exposure to stressors [144]. Therefore, the activation of CB2R and the inhibition of CB1R reduce the behavioral and molecular effects of psychostimulants. Nonetheless, the use of selective CB1R antagonists in humans is limited by the incidence of adverse psychiatric effects, and CB2R appears to be a promising alternative to psychostimulants [145]. However, some recent preclinical studies have suggested that the neutral CB1R antagonist, AM4113, may exert therapeutic and anti-addictive effects; for instance, in addition to preventing heroin self-administration [146], it reduces nicotine and THC self-administration and priming- or cue-induced reinstatement of drug-seeking behavior [147]. Also, AM6527, a novel and promising neutral CB1R antagonist, has been shown to inhibit dose dependently both cocaine and heroin self-administration and cocaine- or heroin-triggered reinstatement of drug-seeking behavior in rats [148]. Furthermore, LY2828360, a G protein-biased CB2R agonist, blocked morphine-induced CPP [149] and partially attenuated naloxone-precipitated opioid withdrawal in morphine-dependent mice [150]. However, randomized controlled clinical trials involving patients suffering from dependence or experiencing withdrawal from psychostimulants are still scarce. Preclinical research on cannabinoid treatments for opioid dependence or withdrawal is somewhat contradictory. THC reduces [113,115] or enhances [112] heroin self-administration, and CB1R antagonism prevents heroin [146] and morphine self-administration and relapse [116], whereas CB1R agonism enhances morphine self-administration [117]. However, a few studies have explored the impact of cannabinoid treatments on opioid dependence and withdrawal [151]. In this way, oral CBD (400 and 800 mg/kg), once daily for three consecutive days, reduced cue-induced craving, assessed according to the visual analog scale for cravings (VAS-C), for anxiety (VAS-A), and the Positive and Negative Affect Schedule, in a sample of 42 patients, between 21 and 65 years of age, which met the DSM-IV classification for opioid dependence [152].

## 8. Conclusions

Several papers exploring the impact of cannabinoid-based therapies on sleep disorders [153] and addictive behaviors [142,154] have been published in the last few years, including original research papers and systematic reviews of preclinical and clinical studies. However, the literature on the impact of cannabinoids on sleep disorders still lacks methodological homogeneity. On the one hand, in preclinical studies in which variables are rigorously controlled, differences in the doses, route of administration, i.e., ICV, IP, and IV, etc., and the period of administration of cannabinergic compounds, the pharmacodynamics of CB1R/CB2R ligands, i.e., their efficacy and potency, are the subject of debate. On the other hand, the methodological heterogeneity in clinical studies, i.e., qualitative vs. objective measures of sleep, in addition to the population studied, i.e., insomniac patients, volunteers with sleep disturbances, abstinent patients in regard to psychostimulants or depressants, makes it difficult to conclude the efficacy of these compounds. Regarding the impact of cannabinoids in the treatment of dependence or withdrawal from psychostimulants and depressants, animal studies point to the efficacy of CB1 antagonists and CB2R agonists to inhibit self-administration and prevent relapse. Yet, clinical trials evaluating CB1R/CB2R-based pharmacotherapy in patients exhibiting addictive behaviors involving psychostimulants or depressants are scarce.

## 9. Future Directions

Although Professor Raphael Mechoulam, the greatest medicinal cannabis researcher, once said, “I believe cannabinoids represent a medical treasure waiting to be discovered”, and despite the fact that some therapeutic effects of cannabinoids, cannabinoid derivatives, and synthetic cannabinoids include the palliation of pain and nausea, and can be used in the treatment of seizure disorders, ischemia, and cerebral trauma [155,156], their potentially deleterious side effects must be evaluated carefully. Although the literature on the role of CB1R antagonists and CB1R-knockout mice in addiction is wide ranging [157], the US Food and Drug Administration rejected rimonabant (SR141716), a CB1R antagonist/inverse agonist, for the treatment of obesity and nicotine dependence, due to its adverse psychiatric effects, including an increased risk of depression, anxiety, and suicidality [158,159]. Also, the systemic acute administration of HU210 (0.05–1 mg/kg), a potent synthetic cannabinoid, and THC (5 mg/kg), the major psychoactive constituent of marijuana, impairs working memory via the long-term depression of the CB1R astroglial-dependent mechanism in regard to CA3–CA1 synapses in mice [160]. Likewise, increased negative emotionality and reduced dopamine sensitivity have been reported in long-term marijuana consumers [94]. Recently, it has been argued that the adverse side effects associated with rimonabant may be related to its inverse agonist profile [161]. In this way, although some recent preclinical studies have suggested the therapeutic potential of the neutral CB1R antagonists, AM4113 and AM657, as well as the G protein-biased CB2R agonist, LY2828360, for the treatment of addiction to opioids [142], THC, and nicotine [147], psychostimulants, and depressants [148,149,150], without apparent adverse side effects, further preclinical research and clinical trials are required. Moreover, sleep abnormalities have been identified in alcohol, cannabis, cocaine, and opiate consumers [162], but studies exploring the potential use of these compounds for treating sleep disorders have not been published.

## Figures and Tables

**Figure 1 pharmaceuticals-18-00266-f001:**
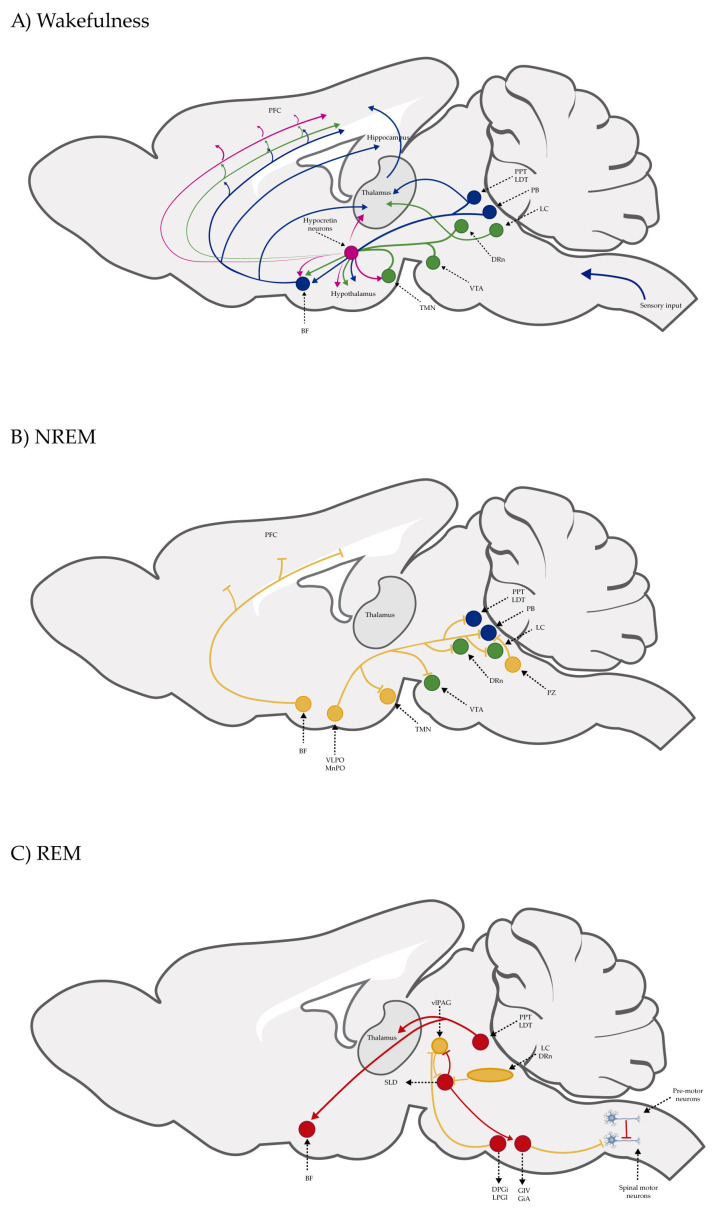
Sagittal schematic perspectives of a rat brain, indicating the neuronal networks responsible for waking, NREM, and REM. (**A**) Circuit of wakefulness regulation. The orexinergic neurons (magenta circles) from the lateral hypothalamus send excitatory projections to noradrenergic neurons in the LC, serotoninergic neurons in the DRn, dopaminergic neurons in the VTA, and histaminergic neurons in the TMN. Also, the LC, PPT, and LDT send projections to the thalamus–cortex. Basal forebrain (BF) neurons also send projections to the cerebral cortex, leading to cortical activation. (**B**) Circuit of NREM sleep regulation. GABAergic neurons in the VLPO and MnPO promote sleep by inhibiting arousal-promoting neurons in the hypothalamus and brain stem (green circles). In addition, the BF includes sleep-promoting neurons (GABAergic neurons) that are projected onto the prefrontal cortex (PFC). Also, GABAergic neurons in the PZ promote sleep by inhibiting PB neurons. Yellow circles: neurons representative of the brain nucleus that promote sleep. (**C**) Circuit of REM sleep regulation. During REM sleep, the vlPAG is inhibited by SLD GABAergic neurons. REM sleep-promoting neurons and REM sleep-suppressing neurons are represented by the red and yellow circles, respectively. The glutamatergic neurons in the SLD produce muscle paralysis during REM sleep via GABAergic/glycinergic neurons in the ventromedial medulla and spinal cord that hyperpolarize motor neurons. GABAergic neurons in the vlPAG and monoaminergic neurons in the LC and DRn inhibit SLD. Conversely, the PPT and LDT cholinergic neurons are permissive in regard to promoting REM sleep (Modified from [48]). Abbreviations: LC: locus coeruleus; DRn: dorsal raphe nucleus; VTA: ventral tegmental area; TMN: tuberomammillary nucleus; PPT: pedunculopontine tegmental nucleus; LDT: laterodorsal tegmental nucleus; VLPO: ventrolateral preoptic nucleus; MnPO: median preoptic nuclei; PB: parabrachial nucleus; PZ: parafacial zone; DPGi: dorsal paragigantocellular reticular nuclei; GiV: ventral gigantocellular reticular nuclei; GiA: alpha gigantocellular reticular nuclei; LPGi: lateral paragigantocellular nuclei; SLD: sublaterodorsal nuclei; vlPAG: ventrolateral periaqueductal gray.

**Table 1 pharmaceuticals-18-00266-t001:** The CB1R and CB2R ligands employed in treating sleep disorders.

Classification	Chemical Structure	Animal Model/Clinical Trial	Route of Administration, Dose, and Period	Main Effects	Ref.
CB1R					
ABD459/Antagonist	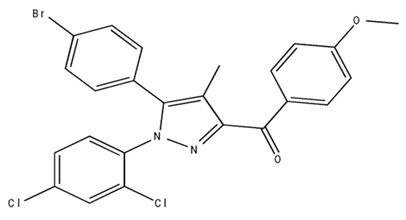	C57Bl/6 mice	IP, 3, 10 and 20 mg/kg, 1 day	Reduced REM sleep, without alterations to NREM and total sleep.	[68]
SR141716A/Antagonist	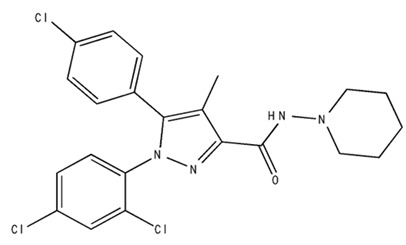	Male Wistar rats	IP, 5, 10 or 20 mg/kg, 1 day	Prevention of the sleep rebound in total sleep-deprived rats in a dose-dependent fashion.	[69]
WIN 55, 212–2/Agonist	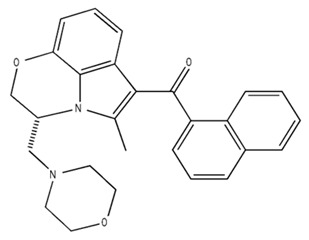	Male Wistar rats	IP, 0.1, 0.3, or 1.0 mg/kg, 14 days	Alterations in sleep patterns; decreased wakefulness and enhanced REM sleep in adult rats.	[70]
WIN 55,212–2 mesylate salt/Agonist	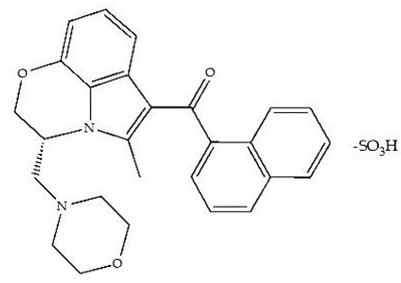	Male Wistar rats	Medial septum MI, 200 nL, 6 h	MI of a CB1R agonist decreased NREM and increased total sleep time with a high percentage of REM sleep, while the MI of a CB1R antagonist decreased NREM and REM sleep.	[71]
LY 320,135/Antagonist	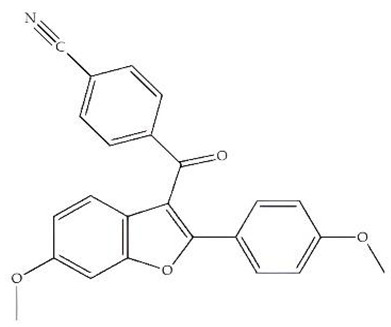				
VD-hemopressin α/Agonist	Val-Asp-Pro-Val-Asn-Phe-Lys-Leu-Leu-Ser-His-OH	Male Sprague-Dawley rats	ICV 6.7, 13.4, and 20.1 nmol, 24 h	NREM sleep enhancement and stabilization.	[72]
Anandamide/Agonist	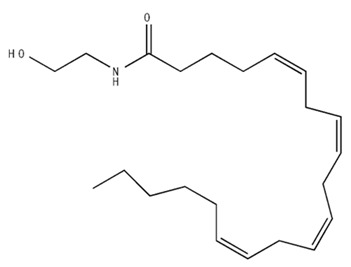	Male Wistar rats	Oral, 50 mg, 8 wk	Improved sleep quality and increased Natural Killer immune cell function.	[73]
Anandamide/Agonist	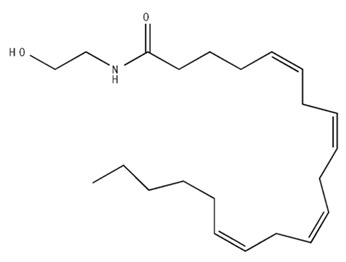	Male Wistar rats	Oral, 20 mg/kg, 21 days	Reduction in anxiety symptoms in sleep-deprived rats.	[74]
Cannabinol (CBN)/Partial agonist	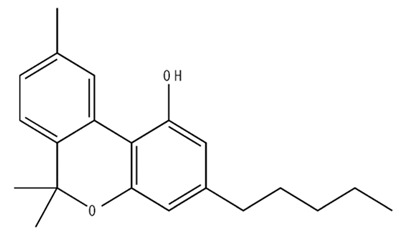	Male Long–Evans rats	IP, 10, 30, and 100 mg/kg, 3 injections, 15 days	Increased total sleep time, also increased REM and NREM sleep. The impact on NREM was similar to the administration of zolpidem.	[75]
CB2R					
Dronabinol/Agonist (synthetic THC)	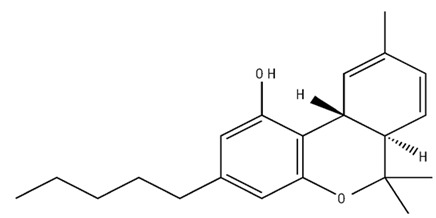	Clinical	Oral, 2.5, 5, and 10 mg, 21 days	Reduction in obstructive sleep apnea.	[76]
Dronabinol/Agonist (synthetic THC)	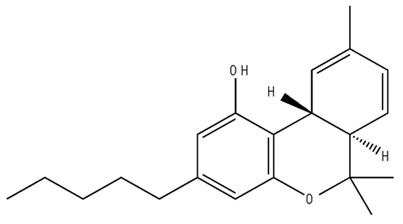	Male Sprague-Dawley rats	IP, 5 mg/kg, and combined 5/5 mg/kg, 6 h	Dronabinol decreased the percent of REM sleep and apnea during sleep, but when combined with the antagonist, the reduction in sleep apnea was reversed.	[77]
AM 251/Antagonist	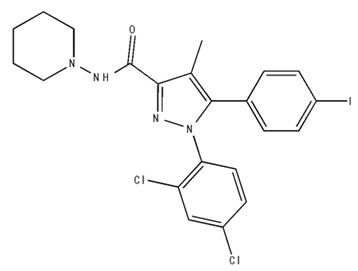				
AM 630/Antagonist	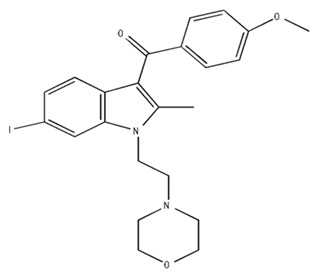				
Cannabidiol/Negative allosteric modulator of CB1R	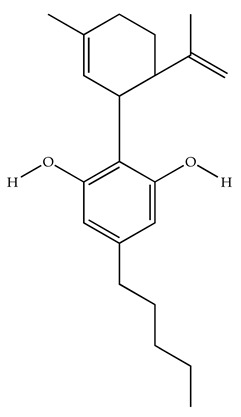	Clinical	Oral, 50 mg, 8 wk	Improved sleep quality and increased Natural Killer immune cell function.	[78]

Abbreviations: Ref: reference; IP: intraperitoneal; wk: weeks; ICV: intracerebroventricular; MI: microinjection. The 2D structure images were obtained from “https://www.chemspider.com (accessed on 6 February 2025)” and “https://pubchem.ncbi.nlm.nih.gov/ (accessed on 6 February 2025)”.

**Table 2 pharmaceuticals-18-00266-t002:** The CB1R and CB2R ligands employed in the treatment of addictive behaviors.

Classification	Chemical Structure	Animal Model/Clinical Trial	Route of Administration, Dose, and Period	Main Effects	Ref.
CB1R					
AM4113/Neutral antagonist	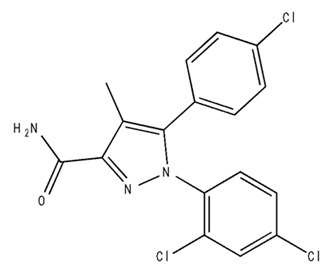	Male Long–Evans and Wistar rats	IP, 1,3, 10 mg/kg, 10 days	Reduced nicotine consumption, decreased motivation for nicotine, and diminished reinstatement of nicotine-seeking behavior.	[129]
SR141716/Antagonist/Inverse agonist	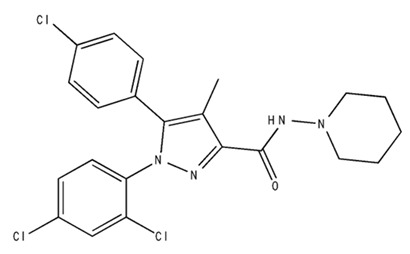	Male C57Bl6 mice	IP, 10 mg/kg, 1 day	Precipitation of withdrawal signs in mice chronically treated with THC or WIN55,212.	[130]
Δ^9^-THC/CB1 and CB2 agonist	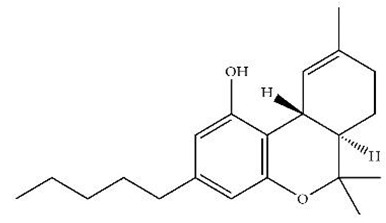	Adult male Long–Evans rats	IP, 3 mg/kg	Biphasic effects mildly enhance BSR at low doses, but inhibit it at higher doses. Pretreatment with AM251 attenuated low dose-enhanced BSR, while AM630 attenuated high dose-inhibited BSR.	[131]
WIN55,212-2 CB1 and CB2/Agonist	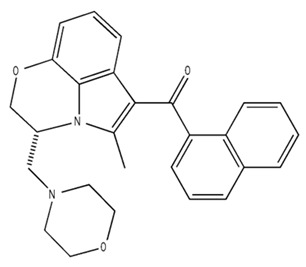				
AM251 CB1/Antagonist	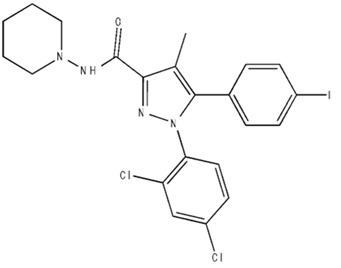				
AM630 CB2/Antagonist	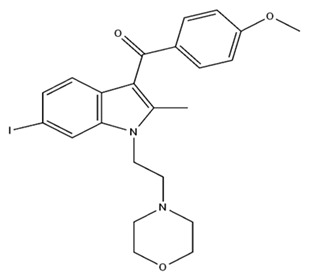				
Δ^8^-THCV/CB1 antagonist and CB2 agonist	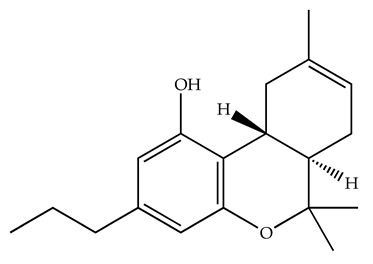	Male alcohol-preferring Wistar rats and male drug-naïve ICR mice	IP, 3 or 10 mg/kg, and 10 or 20 mg/kg	Attenuated intravenous nicotine self-administration and both cue-induced and nicotine-induced relapse to nicotine-seeking behavior in rats. In mice it also significantly attenuated nicotine-induced conditioned place preference and nicotine withdrawal.	[132]
∆^9^-THC/Agonist	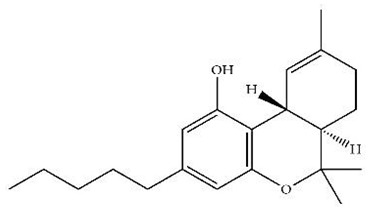	Clinical	Oral, 7.5 and 15 mg	Dampens respond to reward and loss feedback, which may reflect an “amotivational” state.	[133]
PIMSR/Antagonist	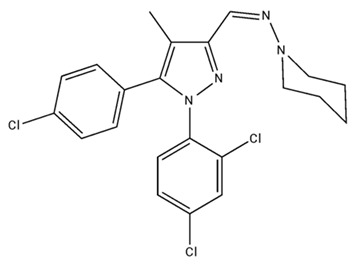	Male Long–Evans rats and male and female wild-type (C57/BL6J) mice	IP, 3, 10 and 30 mg/kg 4 wk	Dose dependently inhibited cocaine self-administration, decreased motivation to seek cocaine under progressive ratio reinforcement, and reduced cue-induced reinstatement of cocaine-seeking behavior.	[134,135]
AM251/Antagonist/Inverse agonist	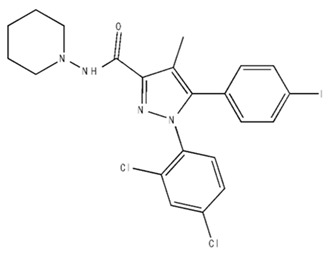	Male Sprague-Dawley rats	IP, 1 mg/kg 2 wk	Attenuated cocaine intake only in rats with a history of stress.	[136]
CB2R					
APD 371/CB2 agonist	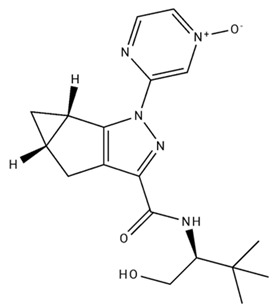	Male Sprague-Dawley rats	Oral 30 mg/kg, time not specified	Generated hyperalgesia without tachyphylaxis in morphine-treated osteoarthritis rats.	[137]
AM1241/Agonist	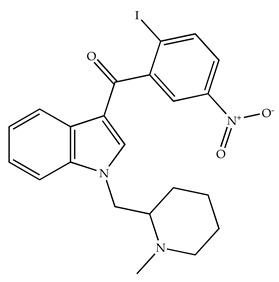	Male C57B6/J mice (6 weeks)	IP 3 or 6 mg/kg, 4 wk	Reduction in anxiety-like behaviors in chronic alcohol-exposed adolescent mice.	[138]
AM1710/Agonist	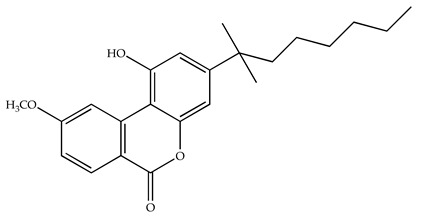	Adult male and female C57Bl6/J mice	IP 25 mg/kg 3 wk	Reversal of morphine tolerance and dependence.	[139]
LY2828360/Agonist	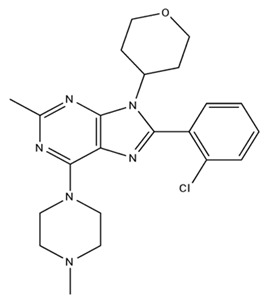				

Abbreviations: Ref.: reference; IP: intraperitoneal; wk: weeks; BSR: brain stimulation reward. The 2D structure images were obtained from “https://www.chemspider.com (accessed on 6 February 2025)” and “https://pubchem.ncbi.nlm.nih.gov/ (accessed on 6 February 2025)”. 2D structure from reference [135].

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
