# Peer review of "Involvement of CB1R and CB2R Ligands in Sleep Disorders and Addictive Behaviors in the Last 25 Years"

_pharmaceuticals, 2025, doi:10.3390/ph18020266_

Round 1
Reviewer 1 Report
Comments and Suggestions for Authors
The authors conducted a review of 150 papers on the role of the cannabinoid system in various physiological functions, with a specific focus on the brain. They examined the effects of agonistic and antagonistic ligands on CB1/ CB2 receptors in relation to therapies for insomnia and addiction. The most pertinent data are summarised in Tables 1 and 2, which are discussed in detail. This paper is relevant for both basic researchers and clinicians.
Reviewer 2 Report
Comments and Suggestions for Authors
- In the present study, the authors provide a comprehensive description of the brain mechanisms of cannabinoids and their effects on sleep disorders and addictive behaviors. This work is both interesting and highly relevant to the field of cannabinoid research. I have a few comments and suggestions to improve the quality of the manuscript, which are outlined below:
-
The terms Marihuana and Marijuana are both acceptable in the literature. I suggest the authors include Marijuana in their search strategy to explore whether additional articles relevant to this review can be identified. Additionally, the authors should incorporate the keyword substance use disorder (SUD) into their search strategy.
-
The authors should consider including The 2018 Farm Bill in the introductory section for context.
-
The manuscript title should clearly indicate the time period covered by the literature search.
-
Page 2, lines 78–79: The statement, “delta-9-tetrahydrocannabinol (THC) and cannabidiol (CBD) have been identified as its main psychoactive compounds [10,11],” requires refinement. CBD is a non-psychoactive compound and should not be described as such.
-
Page 7, line 319: In the sentence, “only AM251 (0.3–1 mg/kg) decreases cue-induced reinstatement,” please specify whether the cue-induced reinstatement is “discrete” or “discriminative.”
-
Table 1: The term “Cannabibol (CBN)/Partial agonist” should be corrected to “Cannabinol (CBN)/Partial agonist.”
-
Table 1 (under Gueye et al., 2016 [109]): The dose “IP, 1,3 10 mg/kg” should be revised to “IP, 1, 3, 10 mg/kg” for consistency and clarity.
-
Page 11, lines 429–430: The statement, “3 consecutive days oral CBD (400 and 800 mg/kg), which is an agonist of CB1R and CB2R, reduces cue-induced craving,” is inaccurate. CBD is not considered an agonist of the CB1 and CB2 receptors. Research indicates that CBD primarily acts as an antagonist or negative allosteric modulator at these receptors. Please revise accordingly.
-
Tables 1 and 2: These tables could be combined for greater clarity. Furthermore, I recommend the authors include 2D structures of the compounds mentioned in Tables 1 and 2, as well as within the text of the manuscript, to enhance the presentation and comprehension of the data.
It can be improved if the authors carefully proofread the entire manuscript.
Reviewer 3 Report
Comments and Suggestions for Authors
This is a good review article that covers a wide range of information about cannabinoid receptors (CB1R and CB2R) and their role in sleep disorders and addictive behavior. The authors clearly explained the introductory part, and then discussed in detail the mechanisms of action, including phytocannabinoids, endocannabinoids and synthetic analogues. The article is well structured, which facilitates its understanding, and includes both preclinical and clinical data. The topic of the article is relevant in light of the growing interest in the therapeutic use of cannabis and its derivatives. Tables help to summarize information and make it more accessible to readers. The article ends with conclusions that summarize the main points and suggest future research directions.
Comments and suggestions for improvement:
· In my opinion, the article lacks an illustration, perhaps some drawings, molecular formulas, etc. Specific examples of synthetic cannabinoids should be provided so that the reader understands what is being discussed. Maybe authors should add some diagrams or illustrations showing the relationship between the different parts of the brain involved in sleep regulation.
· When describing synthetic cannabinoids, it would be useful to clarify which specific compounds were meant by “K2” and “Spice” (if it does not contradict any norms and laws)
· The text contains several typos and grammatical errors that should be corrected.
Reviewer 4 Report
Comments and Suggestions for Authors
This manuscript reviewed the roles of cannabinoid CB1R and CB2R, in sleep disorders and addictive behaviors. Notably, cannabinoids like THC and CBD influence REM and NREM sleep patterns, showing potential for alleviating conditions like insomnia and sleep apnea. However, evidence from human studies remains inconsistent. Additionally, CB1R agonists are associated with enhanced reward-seeking behaviors, while CB1R antagonists and CB2R agonists have shown efficacy in reducing drug-seeking behaviors and relapse in preclinical models. Despite their therapeutic potential, the manuscript highlights the need for rigorous clinical trials. Overall, it is an interesting review and I have a few comments and suggestions to further improve this article.
- The abstract could be improved by including a concise summary of the key findings or gaps identified in the literature.
- Consider condensing the introduction to focus more directly on the research objectives and the significance of the topic.
- Ensure consistent use of terms, such as standardizing "marijuana" instead of "marihuana."
- Some citations are outdated or overly general. Recent studies, particularly those published near and after 2020 (e.g., PMID: 38600154, 26888056, 27493155, 29967454, 36291128, 34566647, 38631564, 39644993, 32896549, 33069159, 31883107, etc.), should be included to better reflect the current state of the field.
- Splitting Table 1 into two tables—one focusing on sleep disorders and the other on substance use disorders (SUDs)—could enhance clarity.
- The discussion section could delve deeper into clinical implications, providing practical recommendations for the use of CB1R/CB2R agonists or antagonists in treating sleep disorders or addiction.
- Adding graphical abstracts or visual summaries to illustrate complex mechanisms or study results would make the content more accessible.
- Refocus the conclusion to summarize the key findings concisely and highlight the most critical directions for future research.
Reviewer 5 Report
Comments and Suggestions for Authors
Pérez-Morales et al. describe mechanisms related to the cannabinoid system in sleep and addiction. Although the topic is generally of interest, there are so many flaws in the text that the manuscript should be rejected.
1. Reading l. 47 – 49 one might think that cannabis regulation has been introduced only recently. As a matter of fact “drugs” have been regulated at the international level since 1912, while cannabis has been specifically regulated since 1925.
2. T.H. Mikuriya has examined the usefulness (?) of cannabis products in human addiction. This author has not been cited at all.
3. The authors repeatedly mention the use of cannabis products in the experimental vs. clinical setting. The reader does not learn for which disorders cannabis products (e.g., nabilone, dronabinol, nabiximols, cannabidiol) are indicated in humans.
4. L. 430: CBD should not be classified as a CB1R and CB2R agonist.
5. The authors have introduced a series of compounds acting on the cannabinoid system. Cannabinoids apart from THC and CBD were not considered but surprisingly appear in the tables (e.g., cannabinol and THCV).
6. A similar situation occurs for CBR agonists (e.g. WIN55,212-2) which although occurring in the tables have not been considered in the part of the manuscript dealing with different sets of compounds acting on the cannabinoid system. On l. 109 – 121 synthetic cannabinoids are discussed although they do not play a role in the rest of the manuscript.
7. The nomenclature used by the authors has to be questioned. Terms like „cannabis agonist“ (Table 1) should be avoided. The relationship between THC and dronabinol has not been explained.
8. Tables have not been prepared carefully. Look at Table 1. Abbreviations are not consistently explained. Cannabinol is misspelled. Doses of drugs and durations of drug treatments are not given consistently. In the line dedicated to ref. 111 the effect of the drugs is not really clear. In the upper part, agonism at both CB1R and CB2R is shared by THC and WIN. However, with respect to the antagonists, AM251 is selective for CB1R and AM630 for CB2R.
9. Tables should be accompanied by a short discussion in the text. This is not the case in this manuscript. Note that in the text ref. up to 100 are discussed in the text whereas ref. 101 through 115 appear in Table 1 only. Mutatis mutandis, the same situation is true for Table 2.
10. I got the impression that the authors want to restrict themselves to agonists and antagonists (l. 148 – 151). This is not really possible since an agonist like THC behaves as a full or partial agonist depending on the model. SR141716, again depending on the model, behaves as an antagonist or inverse agonist.
Minor points:
l. 11: industry … increased???
l. 38: „nonetheless“ is misleading: There is no contradiction between the parts before and after l. 38.
l. 88: There should be a hierarchical order: mammals, birds, reptiles and fishes.
l. 94: I think that the relationship between microglia and immune system should be elaborated.
l. 98: increasing
l. 117: 2-100 times more potent
l. 132: Include nabilone… What do you mean?
l. 148: What does „cannabinergic activity“ mean?
l. 170: „wake“ should be further specified (e.g., wake condition, wake individuals, during wakefulness).
l. 186: increases, decreases
l. 203: Replace „despite“ by „although“.
l. 378: The reader should be referred to Tables 1 and 2 earlier in the manuscript.
l. 401: routinary?
Round 2
Reviewer 4 Report
Comments and Suggestions for Authors
The authors have addressed most of my comments appropriately. I have no additional comments.
Author Response
Thank you very much for your comments.
Reviewer 5 Report
Comments and Suggestions for Authors
In the table ranging from p. 7 to p. 9, no less than 10 chemical structures are wrong: p. 7, lines 1, 2 and 4; p. 8, lines 3, 4, 5, 6 and 7 (twice) and p. 9, line 2. After having seen this table, I stopped checking the rest of the letter to the editor. Errors of that kind must not occur in a journal termed "Pharmaceuticals". The manuscript has to be rejected and an additional revision should not be possible.
